# Advances in Metabolic Bariatric Surgeries and Endoscopic Therapies: A Comprehensive Narrative Review of Diabetes Remission Outcomes

**DOI:** 10.3390/medicina61020350

**Published:** 2025-02-17

**Authors:** Wissam Ghusn, Jana Zeineddine, Richard S. Betancourt, Aryan Gajjar, Wah Yang, Andrew G. Robertson, Omar M. Ghanem

**Affiliations:** 1Department of Internal Medicine, Boston Medical Center, Boston, MA 02118, USA; wissamghusn7@gmail.com; 2Division of Gastroenterology and Hepatology, Department of Medicine, Mayo Clinic, Rochester, MN 55905, USA; 3Department of Colorectal Surgery, Massachusetts General Hospital, Boston, MA 02114, USA; janazeineddine.md@gmail.com; 4Department of Surgery, Endocrine and Metabolic Surgery, Mayo Clinic, Rochester, MN 55905, USA; betancourt.kylerichard@mayo.edu (R.S.B.); aryangajjar0722@gmail.com (A.G.); 5Department of Metabolic and Bariatric Surgery, The First Affiliated Hospital of Jinan University, Guangzhou 510632, China; yangwah@connect.hku.hk; 6Clinical Department of Surgery, University of Edinburgh, Royal Infirmary, Edinburgh EH8 9YL, UK

**Keywords:** diabetes, RYGB, SG, DS, ESG, obesity, balloon

## Abstract

*Background and Objectives*: Type 2 diabetes (T2D), closely associated with obesity, contributes to increased morbidity and mortality due to complications such as cardiometabolic disease. This review aims to evaluate the effectiveness of metabolic and bariatric surgeries (MBS) and endoscopic bariatric therapies (EBTs) in achieving diabetes remission and to examine key predictors influencing remission outcomes. *Materials and Methods*: This review synthesizes data from studies on MBS and EBT outcomes, focusing on predictors for diabetes remission such as preoperative insulin use, diabetes duration, HbA1c, and C-peptide levels. Additionally, predictive scoring systems, including the Individualized Metabolic Surgery (IMS), DiaRem, Advanced-DiaRem, ABCD, and Robert et al. scores, were analyzed for their utility in forecasting remission likelihood. *Results*: Key predictors of T2D remission include shorter diabetes duration, lower HbA1c, and higher C-peptide levels, while prolonged insulin use, and higher insulin doses are associated with lower remission rates. Scoring models like IMS and DiaRem demonstrate that lower scores correlate with a higher likelihood of remission, especially for procedures such as Roux-En-Y gastric bypass (RYGB). RYGB generally shows higher remission rates compared to sleeve gastrectomy (SG), particularly among patients with mild disease severity, while EBTs like ESG and IGBs contribute 5–20% total weight loss (TWL) and moderate glycemic control improvements. *Conclusions*: Both MBS and EBTs are effective for T2D management, with predictive scoring models aiding in individualized patient selection to optimize remission outcomes. Further research to validate these predictive tools across diverse populations could enhance treatment planning for both surgical and endoscopic interventions.

## 1. Introduction

Type-2 diabetes (T2D), a chronic metabolic disorder characterized by hyperglycemia, has become a global health crisis, affecting over 500 million people worldwide and accounting for over 10% of the population [1]. The burden of diabetes is immense, contributing to significant morbidity and mortality due to its association with complications such as cardiovascular disease, neuropathy, nephropathy, and retinopathy [2,3,4]. Managing diabetes is critical to reducing some of these risks and improving the quality of life for millions of individuals. Moreover, T2D is particularly more prevalent in patient with obesity (i.e., BMI > 30 kg/m^2^) [5,6,7]. The connection between obesity and T2D is driven by intricate cellular and physiological processes, including changes in β cell function, alterations in adipose tissue biology, and insulin resistance across multiple organs, all of which can be improved or even normalized through sufficient weight loss [8].

Among the various treatment options, metabolic and bariatric surgeries (MBS) and endoscopic bariatric therapies (EBT) have emerged as highly effective interventions for achieving sustained weight loss and improving glycemic control in patients with T2D [9,10,11,12,13,14]. Importantly, MBS have demonstrated remarkable diabetes remission outcomes in a wide population of patients, irrespective of preoperative BMI [12,15]. Procedures such as Roux-en-Y Gastric Bypass (RYGB), Sleeve Gastrectomy (SG), and the Duodenal Switch (DS) are particularly notable for their profound effects on weight loss and metabolic improvements, including the potential for diabetes remission [16]. EBT including space-occupying intragastric devices (IGDs), aspiration therapy (AT), incisionless anastomosis devices (IADs), tissue apposition devices (TADs; particularly, Endoscopic Sleeve Gastroplasty [ESG]), duodenal-jejunal bypass line (DJBL), and duodenal mucosal resurfacing (DMR) have also been demonstrating promising weight loss and diabetes remission outcomes [17]. However, limited studies (e.g., methodological limitations, small sample sizes) have been conducted to evaluate the effect of EBT on T2D remission.

Understanding the effects of MBS and EBT on diabetes is of paramount importance, given the growing prevalence of obesity and T2D globally [18,19]. As these procedures are increasingly adopted to manage obesity and related metabolic disorders, it is crucial to evaluate their long-term impacts on diabetes outcomes. This review aims to provide a comprehensive summary of the current knowledge on the effects of MBS and EBTs on diabetes, highlighting the significance of these procedures in improving metabolic health. In addition to summarizing the existing literature, we will portray the key factors that influence diabetes remission following these surgeries, including patient-specific parameters such as preoperative diabetic status and scores. The review will assess how these variables, alongside surgical/endoscopic factors, contribute to remission outcomes and long-term glycemic control.

## 2. Bariatric Procedures: Description

### 2.1. Metabolic Bariatric Surgeries (MBS) (Table 1)

RYGB involves creating a small gastric pouch that is surgically connected directly to the jejunum, bypassing the duodenum (Figure 1A) [20,21]. This alteration in anatomy restricts food intake by reducing stomach capacity and limits nutrient absorption due to the bypassed portion of the small intestine. RYGB also significantly affects gut hormone secretion, via food bypassing the duodenum and proximal jejunum. This notably increases glucagon-like peptide-1 (GLP-1) levels, which enhances insulin sensitivity and supports glycemic control (Table 1) [20,22,23]. These combined effects result in substantial metabolic improvements and sustained weight loss [17].

SG involves surgically removing a large portion of the stomach, leaving a narrow, tube-like “sleeve” structure (Figure 1B) [24,25]. This reduction in stomach size restricts food intake by inducing early satiety and also impacts gut hormones, particularly by reducing ghrelin levels, a hormone central to hunger regulation [26,27].

In addition, DS procedure combines elements of SG and intestinal bypass, providing both restrictive and hypoabsorptive effects [28,29,30]. In this procedure, a portion of the stomach is removed to limit food intake, while the intestines are rerouted to reduce caloric and nutrient absorption significantly (Figure 1C) [30].

### 2.2. Endoscopic Bariatric Therapies (EBTs) (Table 1)

IGB is a device, positioned within the stomach, that exerts both mechanical and hormonal effects that lead to decreased calorie intake (Figure 2A) [31,32]. Mechanistically, IGBs physically occupy gastric space, reducing the stomach’s accommodation capacity and slowing the rate at which it empties, which prolongs the feeling of fullness after meals [33,34,35,36,37]. Additionally, the presence of IGBs has been associated with an increase in the secretion of hormones such as GLP-1 and peptide YY (PYY), which further enhance satiety signals and reduce appetite [38].

AT represents a novel approach to weight loss, wherein a gastrostomy tube with a siphon assembly is implanted in the stomach to enable postprandial aspiration of gastric contents (Figure 2B) [39,40,41,42]. By aspirating food 20 min after a meal, the device can remove approximately 30% of the ingested calories before they are absorbed. This immediate calorie reduction is further enhanced by behavioral modifications; patients are encouraged to chew food thoroughly and drink more water with meals, fostering slower eating habits and satiety [43,44,45].

IADs enable surgical connections without traditional incisions, creating a bypass that alters nutrient flow and enhances satiety (Figure 2C) [44]. This rerouting of the digestive tract can reduce the absorption of calories and nutrients, promoting weight loss and metabolic improvement [17,44].

TADs, including techniques like ESG, are advanced endoscopic bariatric therapies that reduce gastric volume by creating durable tissue folds within the stomach [46] (Figure 2D). ESG, in particular, uses suturing to form a tubular, sleeve-like structure, significantly restricting stomach capacity and thereby enhancing early satiety with smaller food intake. This reduction in gastric volume achieved by TADs and ESG effectively limits caloric intake, supporting sustainable weight management and improved metabolic health outcomes [17].

The DJBL bypasses the duodenum and a portion of the jejunum, facilitating weight loss through mechanisms that include reduced caloric absorption and increased hormonal changes that regulate appetite and glucose metabolism (Figure 2E). This device has shown promising results in studies, with significant weight loss and improved glycemic control, including remission of diabetes in many patients [17]. However, adverse events, such as liver abscess formation, necessitate careful patient selection and monitoring [17].

DMR is an innovative endoscopic bariatric therapy that employs hydrothermal ablation to rejuvenate the duodenal mucosa (Figure 2F) [47,48,49,50,51]. This process promotes mucosal regeneration, which is thought to restore normal signaling pathways between the intestine and liver, thereby improving insulin sensitivity and enhancing glycemic control in patients with metabolic disorders [17,47,48,49,50,51].

### 2.3. Bariatric Procedures: Weight Loss and Diabetes Remission

In a systematic review and meta-analysis of RCTs, RYGB demonstrated a higher rate of diabetes remission in the first year compared to SG, with an initial remission rate of approximately 63% for RYGB versus 52% for SG, corresponding to a relative risk of 1.21 in favor of RYGB [52]. However, by the 3- to 5-year follow-up, remission rates converge, with both RYGB and SG demonstrating similar efficacy in sustaining diabetes remission, showing approximately 50% remission for both procedures. The greater weight loss observed with RYGB, averaging 25% total body weight loss (TWL) compared to 20% with SG, may contribute to its short-term superiority in glycemic control, while both procedures offer durable benefits for weight management and glycemic outcomes in T2D over the longer term [52].

In another RCT, 60 patients with T2D were assigned to medical therapy, RYGB, or DS. After two years, diabetes remission was achieved in 0% of the medical group, 75% of the RYGB group, and 95% of the DS group (*p* < 0.05), based on fasting glucose < 100 mg/dL and HbA1c < 6.5% without pharmacotherapy [53]. In fact, multiple studies have also shown that DS is the most effective MBS in achieving diabetes remission, with studies reporting remission rates close to 95%, sustained over long-term follow-ups [53,54]. RYGB also demonstrates high remission rates, particularly within the first two years, achieving around 60–75% remission in patients with recent diabetes diagnoses [53,55]. SG shows relatively lower remission rates, with approximately 23–47% remission reported at one year, though rates tend to decline over time [56,57].

In addition, amongst the three major MBS, DS has been shown to achieve the greatest weight loss. In a study analyzing 73,702 patients from the Bariatric Outcomes Longitudinal Database who underwent SG, DS, or RYGB, DS resulted in a greater reduction in BMI by 5.3 units, and RYGB by 2.2 units, compared to SG [58]. In terms of comorbidity resolution, RYGB was most effective for gastroesophageal reflux disease (GERD) with an odds ratio (OR) of 1.88, while DS showed superior outcomes for hypertension (OR = 2.12) and diabetes mellitus (OR = 2.53) [59]. The high efficacy of DS stems from its combination of restrictive and hypo-absorptive mechanisms. However, it’s use is limited by the fact that it is a more complicated procedure, making it commonly reserved for patients who require more aggressive interventions [60,61].

Less invasive procedures (i.e., EBTs) have lower overall weight loss and diabetes remission efficacy compared to surgical interventions. EBTs offer diverse mechanisms to promote weight loss. IGBs work by delaying gastric emptying, enhancing gastric accommodation, and increasing GLP-1 and PYY secretion. They achieve approximately 5–15% TWL at 12 months, with most devices intended for 6-month placement to 12 months [17]. Tissue apposition devices, like ESG, primary obesity surgery endoluminal, and endoscopic gastric plication, promote gastric restriction, reduced accommodation, and increased cholecystokinin, yielding around 7–20% TWL [17,62]. A network meta-analysis evaluated the comparative efficacy and safety of EBTs, including ESG and various IGBs, by analyzing randomized controlled trials up to May 2023. The study found that all EBTs resulted in significantly greater %TWL compared to controls, with ESG and the Spatz3 adjustable balloon demonstrating superior %TWL at approximately 12 months post-procedure compared to the Orbera^®^ balloon [63].

AspireAssisst^®^, an aspiration therapy device, allows patients to drain undigested food and includes lifestyle modification, showing around 15–20% TWL over 12 months [17]. Duodenal mucosal resurfacing devices, including Revita^®^ and Diagone^®^, have unclear mechanisms and limited weight loss outcomes. Endoscopic anastomosis devices, such as Incisionless Magnetic Anastomosis System (IMAS^®^), Magnamosis^®^, and EasyByPass^®^, create anastomoses to enhance passage of partially digested food to the distal ileum, increasing GLP-1 and PYY secretion, with IMAS^®^ showing 10–15% TWL and others presenting limited data [17]. Lastly, duodenal-jejunal bypass liners like EndoBarrier^®^ and ValenTx^®^ allow undigested nutrients to bypass the proximal intestine, leading to GLP-1 and PYY increases and achieving around 15–20% TWL with placement durations from 3 to 12 months. Each device category offers a unique mechanism and weight loss potential, adding flexibility in patient-specific EBT selection [17].

IGBs like Orbera^®^ and the Bioenteric^®^ Balloon have shown humble results in reducing obesity-related comorbidities, including T2D. A case series involving 143 patients reported a reduction in diabetes prevalence from 32.6% to 20.9% at six months, with diabetes incidence remaining low even a year post-treatment [63]. The AspireAssist^®^ device, a form of aspiration therapy, also demonstrated diabetes management benefits in the PATHWAY trial, which showed significant reductions in glycated hemoglobin (HbA1c) alongside weight loss (−0.36% relative to 5.7% baseline, *p* < 0.0001) [64]. The DJBS, which bypasses the proximal intestine and increases GLP-1 and PYY secretion, yielded an additional 1% decrease in HbA1c over controls in a meta-analysis including 105 patients, suggesting a strong potential for diabetes remission [65]. Additionally, the IMAS^®^, designed to mimic the duodenal switch, led to HbA1c reductions of 1.9% in patients with diabetes and 1.0% in patients with patients with prediabetes, though long-term remission data are limited [66]. Moreover, in patients who underwent ESG, HbA1c significantly decreased in the overall cohort from 6.1 ± 1.1% to 5.5 ± 0.48% and in patients with diabetes or prediabetes from 6.6 ± 1.2% to 5.6 ± 0.51% after 12 months [67].

## 3. Mechanisms Underlying Diabetes Remission

### 3.1. Weight Loss-Dependent Mechanisms

Calorie restriction in individuals helps reduce fat accumulated around specific organs, such as the liver and pancreas [68]. Excessive fat around the pancreas can disrupt blood glucose regulation, causing spikes in blood glucose levels [69]. Through calorie-restricted diets and weight loss, fat deposits in these organs decrease, resulting in better glycemic control and enhanced regulation of blood glucose [69]. Additionally, weight loss from calorie restriction can alleviate mechanical pressure on peripheral tissues, thereby improving insulin sensitivity in skeletal muscle, a key area for overall glucose homeostasis [70]. Studies have shown that very low-calorie diets and calorie restriction can improve blood glucose by reducing glucose and lipid toxicity, decreasing ectopic fat deposits, enhancing insulin sensitivity, and lowering inflammation [71]. Fat accumulation can impair β-cell function and insulin release, while adipose tissue acts as an endocrine organ that produces proinflammatory cytokines, such as TNF-α and IL-6, which alter insulin function [72].

Reducing fat accumulation around these organs also improves glucose output efficiency and glycemic control. As fat around visceral organs decreases, adipokine secretion, including adiponectin, improves while proinflammatory cytokines, such as TNF-α and IL-6, decrease [72]. Furthermore, weight reduction via calorie restriction or bariatric surgery lowers the production of these cytokines, increasing insulin sensitivity. Reduced levels of cytokines, along with inflammatory markers like C-reactive protein, highlight the role of weight loss in diabetes remission through improved glucose absorption and insulin function [73].

Moreover, hepatic insulin resistance due to fatty liver infiltration can exacerbate hyperglycemia. Bariatric surgery helps reduce hepatic glucose production and lower endogenous glucose levels [74]. Additionally, bariatric interventions decrease free fatty acids (FFAs) and adipose tissue levels; high FFAs can impair insulin signaling, prevent hepatic glucose suppression, and reduce glucose absorption in muscles, leading to hyperglycemia [75]. Weight loss, achieved via bariatric surgery, thus decreases fatty acids and lipotoxicity, enhancing insulin sensitivity and allowing patients to achieve stable glucose levels and improved insulin function [76].

### 3.2. Weight Loss-Independent Mechanisms

Various endoscopic and bariatric procedures can influence gut hormones, contributing to diabetes remission. These procedures often increase levels of glucagon-like GLP-1 and PYY, hormones integral to appetite regulation and glucose metabolism. For instance, RYGB and SG raise GLP-1 and PYY levels, promoting satiety, delaying gastric emptying, increasing insulin secretion, and decreasing glucagon production, improving glycemic control even without significant weight loss [77,78,79,80,81]. Additionally, these procedures lower ghrelin, the “hunger hormone”, by resecting part of the stomach where ghrelin is produced, thereby reducing appetite [22].

Bile acids play a crucial role in glucose metabolism as signaling molecules activating receptors like TGR5 and FXR. Following bariatric surgery, increased bile acid levels in the bloodstream stimulate TGR5, enhancing GLP-1 secretion, a hormone that supports proper insulin secretion and blood glucose regulation [82,83]. FXR, activated by bile acids in the liver and intestines, helps to reduce hepatic glucose production and increase insulin sensitivity, further contributing to glucose homeostasis [83]. Additionally, altered bile acid circulation post-surgery impacts the gut-liver axis, enhancing metabolic control by modulating insulin sensitivity and glucose levels independently of weight loss [84]. Studies have shown that these changes in bile acid metabolism after procedures like RYGB can lead to improved insulin sensitivity and β-cell function, both of which are essential for long-term glycemic control [84].

Changes in gut microbiome composition also contribute to metabolic improvements. Obesity and T2D are associated with reduced microbial diversity, but surgical procedures can increase beneficial bacteria such as Akkermansia muciniphila and Bifidobacterium species, which reduce inflammation, strengthen gut barriers, and enhance insulin sensitivity [85,86,87]. These microbial shifts result in higher production of short-chain fatty acids (SCFAs), such as butyrate, which supports gut health, reduces systemic inflammation, and promotes glucose regulation [88]. The gut-brain axis also plays a role, as diet, medications, and surgery influence gut microbiota, affecting appetite, hormone regulation, and insulin sensitivity independently of weight loss [87,89].

## 4. Predictors of Diabetes Remission

Preoperative patient characteristics, such as disease duration, fasting glucose, insulin use, HbA1c, and C-peptide levels, are commonly assessed to understand T2D progression [90,91].

Longer disease duration is consistently associated with lower remission rates, as prolonged exposure to hyperglycemia and insulin resistance can lead to irreversible pancreatic beta-cell dysfunction, making diabetes harder to reverse [92]. Additionally, the presence of vascular complications, both microvascular and macrovascular, reflects advanced diabetes severity and further reduces remission likelihood, with one study noting an odds ratio (OR) of 2.72 for achieving remission in patients without vascular complications compared to those with such diseases [93]. Preoperative insulin use also negatively predicts remission, as reliance on insulin indicates a more advanced disease stage and reduced endogenous insulin production capacity. This decreases the chances of achieving sustained glycemic control post-surgery [94]. Moreover, insulin dosage itself is a critical factor; patients on higher insulin doses experience significantly lower remission rates, emphasizing the dose-dependent impact of insulin on diabetes remission outcomes [91].

Insulin sensitivity, insulin resistance, and beta-cell function significantly influence T2D progression and treatment response. One study demonstrated that MBS improved insulin sensitivity, reduced insulin resistance (i.e., measured by HOMA-IR), and increased beta-cell compensatory capacity (i.e., measured by the Disposition Index [DI]) in patients with and without T2D. Patients with lower preoperative DI showed less improvement in beta-cell glucose sensitivity and insulin secretion, leading to lower remission rates at both 4 and 18 months (0% remission in low DI vs. 57% in high DI at 4 months; 38% in low DI vs. 71% in high DI at 18 months) [95]. However, another study found that HOMA-IR was not significantly predictive of T2D remission (*p* = 0.961) [96]. While these metrics provide insight into the effects of MBS on insulin response and beta-cell function, their utility in predicting surgical outcomes remains under investigation.

To account for variability among predictors, scoring systems have been developed to assist in forecasting diabetes remission post-MBS. The Individualized Metabolic Surgery (IMS) score, DiaRem, advanced DiaRem (ad-DiaRem), ABCD and Robert et al. score incorporate combinations of predictors such as diabetes duration, HbA1c, and insulin use [97]. The IMS score components involve diabetes duration, number of diabetes medications, insulin use, and HbA1c. The scores are divided into three categories of mild (≤25), moderate (26–95) and severe (>95), with a higher score indicating a more severe disease [97]. Findings are consistent even among patients with severe obesity (i.e., BMI > 50 Kg/m^2^), with AUC-ROC of 0.79 [91]. In a retrospective analysis including 20 patients with T2D who underwent ESG, patients with higher IMS scores, indicating more severe disease, were significantly less likely to achieve DR, with remission rates of 60% in mild, 45.5% in moderate, and 0% in severe categories. The findings suggest that the IMS score may be a useful predictor for DR outcomes post-ESG, guiding individualized treatment planning for obesity and diabetes [98].

Moreover, the DiaRem score ranges from 0 to 22 and incorporates HbA1c, insulin use, age, and other diabetes medications into the scoring system [99]. The ad-DiaRem ranges from 0 to 21 and, in addition to the components in DiaRem, incorporates a number of diabetes medications and diabetes duration [100]. In both scoring systems, a lower score is associated with a greater probability of T2D remission.

The ABCD model is a scoring system developed to predict diabetes remission after MBS by assessing critical factors related to patient characteristics and diabetes status. This model includes four main components: Age, BMI, C-peptide levels, and Duration of diabetes (hence, “ABCD”). Age is included as younger patients tend to have better remission outcomes, while BMI accounts for the obesity severity influencing diabetes improvement [101]. As discussed previously, C-peptide levels indicate endogenous insulin production and beta-cell function, with higher levels generally predicting better remission potential. Also, diabetes duration reflects the progression of the disease, with shorter durations associated with a higher probability of remission due to less beta-cell exhaustion.

The Robert et al. score combines factors such as BMI, diabetes duration, HbA1c, fasting glucose, and insulin use into a scoring system ranging from 0 to 5, where higher scores correlate with a higher likelihood of diabetes remission [102]. However, its effectiveness in predicting remission may be limited for patients undergoing SG [91]. One study demonstrated an overall AUC-ROC of 0.75, with no significant differences between procedural subgroups, though remission rates were markedly higher for RYGB compared to SG at specific scores: 68% vs. 38% for a score of 3, 77% vs. 50% for a score of 4, and 100% vs. 71% for a score of 5 [90].

In addition to highlighting the association between diabetes severity and T2D remission, it is critical to underscore the importance of recommending MBS early, particularly for patients with obesity and T2D. Early surgical intervention, even before the development of severe obesity or advanced diabetes, may maximize the chances of remission by preserving pancreatic beta-cell function and mitigating the deleterious effects of chronic hyperglycemia [92]. For example, evidence also suggests that patients with a lower IMS score, less severe diabetes, exhibit significantly better remission rates [93]. Moreover, patient with has not yet developed vascular diseases related to T2D, have higher diabetes remission rates compared to patient with vascular complications [103]. By prioritizing MBS in this population, long-term metabolic and glycemic outcomes can be optimized, while the burden of diabetes-related complications is reduced.

## 5. Limitations

This narrative review has several limitations that warrant consideration. First, the heterogeneity of the included studies in terms of populations, methodologies, and follow-up durations may impact the generalizability of the findings. Second, while the analysis highlights the efficacy of both MBS and EBT in diabetes remission, direct head-to-head comparisons between these interventions remain limited. Third, the reliance on predictive scoring systems, such as IMS and DiaRem, may not fully account for all patient-specific variables influencing remission outcomes. Finally, the lack of long-term follow-up data for EBTs limits the ability to assess the durability of their effects compared to established surgical interventions. Future prospective studies with standardized designs and long-term outcome assessments are needed to validate and expand upon these findings.

## 6. Conclusions

This review underscores the value of MBS and EBTs as impactful interventions for achieving T2D remission, particularly in patients with obesity who face elevated risks from diabetes-related complications. Through a comprehensive analysis of predictive factors, including diabetes duration, HbA1c, C-peptide levels, and insulin use, this study highlights how patient-specific characteristics can influence remission outcomes. Scoring systems such as the IMS, DiaRem, Advanced-DiaRem, ABCD, and Robert et al. scores provide valuable frameworks for identifying candidates most likely to benefit from these interventions, enabling personalized treatment planning that enhances diabetes remission potential. Given the evidence supporting better outcomes of diabetes remission with MBS and EBT in both younger patients and those with diabetes of a shorter duration, this review supports increased and early access to MBS and EBT for patients, to give them the opportunity for a successful outcome of their therapy.

Future research should focus on refining these predictive models across diverse patient populations and further exploring the long-term metabolic effects of MBS and EBT to optimize outcomes. By aligning patient characteristics with suitable interventions, healthcare providers can better address the global burden of T2D, ultimately improving patient quality of life and reducing healthcare costs associated with diabetes management.

## Figures and Tables

**Figure 1 medicina-61-00350-f001:**
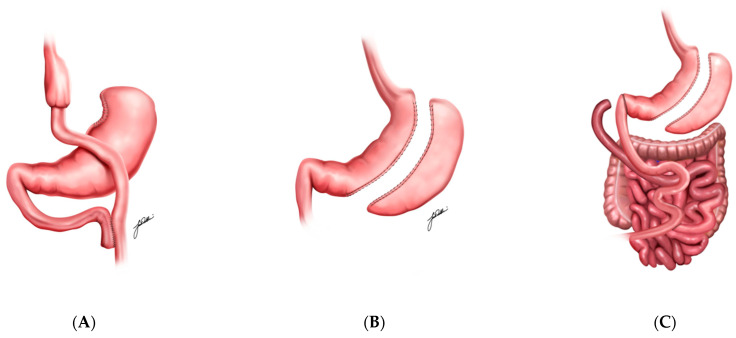
Roux-En-Y Gastric Bypass (**A**), Sleeve Gastrectomy (**B**), and Duodenal Switch (**C**) illustrations.

**Figure 2 medicina-61-00350-f002:**
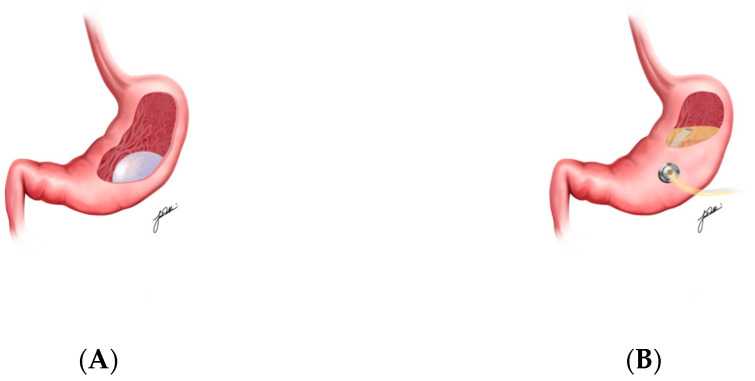
Intragastric balloon (**A**), Aspiration Therapy (**B**), Incisionless Anastomosis Devices (**C**), Endoscopic Sleeve Gastroplasty (**D**), Duodenal-Jejunal Bypass Liner (**E**), and Duodenal Mucosal Resurfacing (**F**) illustrations.

**Table 1 medicina-61-00350-t001:** Mechanism of Action and weight loss outcomes of bariatric procedures.

Procedure/Intervention	Illustration	Mechanism of Action	Weight Loss (%TWL)
Roux-en-Y Gastric Bypass	** 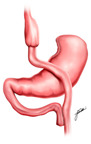 **	Creating a small gastric pouch and bypasses part of the small intestine, increasing GLP-1 secretion and reducing caloric absorption	~25%
Sleeve Gastrectomy	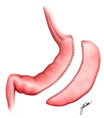	Removing a large portion of the stomach, reducing ghrelin levels and inducing satiety	~20%
Duodenal Switch	** 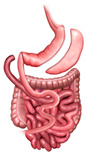 **	Combining restrictive and malabsorptive components, rerouting the intestines to limit caloric absorption and enhance GLP-1 secretion	~35%
Endoscopic Sleeve Gastroplasty	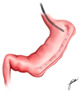	Reducing gastric volume through endoscopic suturing, promoting satiety and reduced calorie intake	~7–20%
Intragastric Balloons	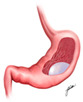	Occupying gastric space, slowing gastric emptying, increasing GLP-1 secretion, and reducing appetite	~5–15%
Aspiration Therapy	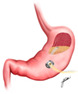	Allowing postprandial aspiration of gastric contents, reducing caloric intake	~15–20%
Duodenal-Jejunal Bypass Liners	** 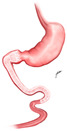 **	Bypassing proximal intestine, enhancing GLP-1 and PYY secretion, reducing calorie absorption	~7–20%
Incisionless Anastomosis Devices	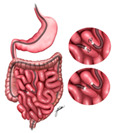	Creating anastomoses to enhance passage of partially digested food to the distal ileum, increasing GLP-1 and PYY secretion	~10–15%
Duodenal Mucosal Resurfacing	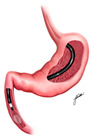	Promoting mucosal regeneration, restoring normal signaling pathways between the intestine and liver, improving insulin sensitivity and enhancing glycemic control	Limited data

Abbreviations used: %TBWL: Percentage of total body weight loss.

## Data Availability

No new data were created or analyzed in this study.

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
