# Peer review of "Advances in Metabolic Bariatric Surgeries and Endoscopic Therapies: A Comprehensive Narrative Review of Diabetes Remission Outcomes"

_medicina, 2025, doi:10.3390/medicina61020350_

Round 1

Reviewer 1 Report

Comments and Suggestions for Authors

General comment

The manuscript presents an interesting research question and gathers relevant information about the remission of type 2 diabetes associated with bariatric and metabolic surgery. Some comments follow to improve the quality of the paper. 

Specific comments

Page 2, lines 59-64. The authors should make it clear that we have few studies, with important methodological limitations and relatively small sample sizes, on the effects of EBT on the remission of type 2 diabetes. 

Page 3, line 94. Replace “malabsorptive” with “hypoabsorptive”, following the most recent terms adopted by IFOS. 

Table 1. Add a footnote with the meaning of “%TWL”. Still, regarding this element, I suggest that the authors review the expected weight loss values with each MBS and EBT technique. I also suggest that the authors add references to weight loss estimates for each of the methods described in this table. 

Page 8, line 200. Replace “P” with “p” as recommended in statistical analysis guidelines.  

Page 8, line 237. The authors report “whether through calorie restriction or bariatric surgery”, but calorie restriction is present in bariatric and metabolic surgery since there is a reduction in food intake and consequently calorie restriction. I suggest that the authors use another term to refer to non-surgical clinical treatment. 

Page 9, lines 273-279. In this excerpt, the authors discuss the impact of disease duration and DM2 remission. I think it is worth adding a short section on the importance of recommending MBS early, not only in severe cases of obesity but especially in patients with obesity and DM2. 

Page 10. Before the conclusion, the authors should include a paragraph commenting on the limitations of this study. 

Minor comments: 

- Figure 2 and Table 1 should be presented after the description of the EBT methods. Currently, table 1 appears even before it is cited in the text. 

Author Response

Comment 1: Page 2, lines 59-64. The authors should make it clear that we have few studies, with important methodological limitations and relatively small sample sizes, on the effects of EBT on the remission of type 2 diabetes. 

Response: We thank our reviewer for this comment. We added the following to the manuscript: 

"However, limited studies (e.g., methodological limitations, small sample sizes) have been conducted to evaluate the effect of EBT on T2D remission."

Comment 2: Page 3, line 94. Replace “malabsorptive” with “hypoabsorptive”, following the most recent terms adopted by IFOS. 

Response: Adjusted as requested.

Table 1. Add a footnote with the meaning of “%TWL”. Still, regarding this element, I suggest that the authors review the expected weight loss values with each MBS and EBT technique. I also suggest that the authors add references to weight loss estimates for each of the methods described in this table. 

Response: Added footnote with %TWL meaning as requested. Also, added references for all weight loss estimated in the table.

Comment 3: Page 8, line 200. Replace “P” with “p” as recommended in statistical analysis guidelines.  

Response: Adjusted as requested.

Comment 4: Page 8, line 237. The authors report “whether through calorie restriction or bariatric surgery”, but calorie restriction is present in bariatric and metabolic surgery since there is a reduction in food intake and consequently calorie restriction. I suggest that the authors use another term to refer to non-surgical clinical treatment. 

Response: We agree with the reviewer. We adjusted accordingly:

"Weight loss, achieved via bariatric surgery, thus decreases fatty acids and lipotoxicity, enhancing insulin sensitivity and allowing patients to achieve stable glucose levels and improved insulin function" 

Comment 5: Page 9, lines 273-279. In this excerpt, the authors discuss the impact of disease duration and DM2 remission. I think it is worth adding a short section on the importance of recommending MBS early, not only in severe cases of obesity but especially in patients with obesity and DM2. 

Response: Thank you for your thoughtful feedback. We appreciate the suggestion to emphasize the importance of recommending metabolic and bariatric surgery early in patients with obesity and type 2 diabetes mellitus. Below is the proposed addition to the manuscript:

" In addition to highlighting the association between diabetes severity and T2D remission, it is critical to underscore the importance of recommending MBS early, particularly for patients with obesity and T2D. Early surgical intervention, even before the development of severe obesity or advanced diabetes, may maximize the chances of remission by preserving pancreatic beta-cell function and mitigating the deleterious effects of chronic hyperglycemia [106]. For example. evidence also suggests that patients with a lower IMS score, less severe diabetes, exhibit significantly better remission rates [107]. Moreover, patient with has not yet developed vascular diseases related to T2D, have higher diabetes remission rates compared to patient with vacular complications [108]. By prioritizing MBS in this population, long-term metabolic and glycemic outcomes can be optimized, while the burden of diabetes-related complications is reduced."

Comment 6: Page 10. Before the conclusion, the authors should include a paragraph commenting on the limitations of this study. 

Response: We have added the requested section as per the reviewer’s suggestion:

"Limitations

This narrative review has several limitations that warrant consideration. First, the heterogeneity of the included studies in terms of populations, methodologies, and follow-up durations may impact the generalizability of the findings. Second, while the analysis highlights the efficacy of both MBS and EBT in diabetes remission, direct head-to-head comparisons between these interventions remain limited. Third, the reliance on predictive scoring systems, such as IMS and DiaRem, may not fully account for all patient-specific variables influencing remission outcomes. Finally, the lack of long-term follow-up data for EBTs limits the ability to assess the durability of their effects compared to established surgical interventions. Future prospective studies with standardized designs and long-term outcome assessments are needed to validate and expand upon these findings."

Comment 7: Figure 2 and Table 1 should be presented after the description of the EBT methods. Currently, table 1 appears even before it is cited in the text.

Response: We agree with our reviewer. However, we have added the Table and Figures at the end of our manuscript. We kindly ask the editorial team to change the location of the table and figure to be presented after the respective citation.

Reviewer 2 Report

Comments and Suggestions for Authors

The manuscript of “Advances in Metabolic Bariatric Surgeries and Endoscopic Therapies: A Comprehensive Narrative Review of Diabetes Remission Outcomes” by Wissam Ghusn and co-authors aims to summarize the novel data on the therapeutic effects of metabolic and bariatric surgeries (MBS) and endoscopic bariatric therapies (EBTs) on diabetes and to evaluate the significance of these procedures in improving metabolic health. The authors highlight the key factors that influence diabetes remission following these surgeries, including preoperative diabetic status and scores. The review validates these variables as predictive tools for assessing remission outcomes and long-term glycemic control.

The relevance of the research topic is due to the sharp increase in the number of type 1 and 2 diabetes mellitus and other metabolic disorders throughout the world and the urgent need to find new approaches for their effective management.

The review is interesting and quite detailed. In general, the manuscript is a detailed review of the current understanding of the effectiveness of metabolic and bariatric surgeries and endoscopic bariatric therapies in achieving diabetes remission, and it supports increased and early access to these surgeries for patients with obesity who face elevated risks from diabetes-related complications.

The manuscript is well written and illustrated. The title, abstract, and keywords correspond to the content of the manuscript. The Introduction fully reflects the current state of the issue under study. The purpose of the work is clearly stated in the Introduction. The description of the data analysis is sufficiently detailed. The authors' assumptions and conclusions are sufficiently substantiated.

The manuscript covers a large amount of literature data (105 sources) and makes a significant contribution to the systematization of knowledge about diabetes and metabolism disorders. The authors have cited a large number of research articles, a significant portion of which have been published over the last five years. Schematic illustrations enhance understanding of the pathological changes in lipid metabolism described in the review. Figures are presented in sufficient quantity and clearly reflect the ideas of the study. The quality of the figures meets the requirements. Overall, the research was conducted at a high level, and the manuscript can be accepted in its current form.

Author Response

Comment 1: 

The manuscript of “Advances in Metabolic Bariatric Surgeries and Endoscopic Therapies: A Comprehensive Narrative Review of Diabetes Remission Outcomes” by Wissam Ghusn and co-authors aims to summarize the novel data on the therapeutic effects of metabolic and bariatric surgeries (MBS) and endoscopic bariatric therapies (EBTs) on diabetes and to evaluate the significance of these procedures in improving metabolic health. The authors highlight the key factors that influence diabetes remission following these surgeries, including preoperative diabetic status and scores. The review validates these variables as predictive tools for assessing remission outcomes and long-term glycemic control.

The relevance of the research topic is due to the sharp increase in the number of type 1 and 2 diabetes mellitus and other metabolic disorders throughout the world and the urgent need to find new approaches for their effective management.

The review is interesting and quite detailed. In general, the manuscript is a detailed review of the current understanding of the effectiveness of metabolic and bariatric surgeries and endoscopic bariatric therapies in achieving diabetes remission, and it supports increased and early access to these surgeries for patients with obesity who face elevated risks from diabetes-related complications.

The manuscript is well written and illustrated. The title, abstract, and keywords correspond to the content of the manuscript. The Introduction fully reflects the current state of the issue under study. The purpose of the work is clearly stated in the Introduction. The description of the data analysis is sufficiently detailed. The authors' assumptions and conclusions are sufficiently substantiated.

The manuscript covers a large amount of literature data (105 sources) and makes a significant contribution to the systematization of knowledge about diabetes and metabolism disorders. The authors have cited a large number of research articles, a significant portion of which have been published over the last five years. Schematic illustrations enhance understanding of the pathological changes in lipid metabolism described in the review. Figures are presented in sufficient quantity and clearly reflect the ideas of the study. The quality of the figures meets the requirements. Overall, the research was conducted at a high level, and the manuscript can be accepted in its current form.

Response 1: We truly thank our reviewer for the kind feedback. We appreciate all the comments.
